# Development of Second Prototype of Twin-Driven Magnetorheological Fluid Actuator for Haptic Device

**DOI:** 10.3390/mi15101184

**Published:** 2024-09-25

**Authors:** Takehito Kikuchi, Asaka Ikeda, Rino Matsushita, Isao Abe

**Affiliations:** 1Faculty of Science and Technology, Oita University, Oita 870-1192, Japan; abe-isao@oita-u.ac.jp; 2Graduate School of Engineering, Oita University, Oita 870-1192, Japan; sonata1999ask@gmail.com (A.I.); v24e6016@oita-u.ac.jp (R.M.)

**Keywords:** actuator, magnetorheological fluid, haptic device, mechanical design

## Abstract

Magnetorheological fluids (MRFs) are functional fluids that exhibit rapid and reproducible rheological responses to external magnetic fields. An MRF has been utilized to develop a haptic device with precise haptic feedback for teleoperative surgical systems. To achieve this, we developed several types of compact MRF clutches for haptics (H-MRCs) and integrated them into a twin-driven MRF actuator (TD-MRA). The first TD-MRA prototype was successfully used to generate fine haptic feedback for operators. However, undesirable torque ripples were observed due to shaft misalignment and the low rigidity of the structure. Additionally, the detailed torque control performance was not evaluated from both static and dynamic current inputs. The objective of this study is to develop a second prototype to reduce torque ripple by improving the structure and evaluating its static and dynamic torque performance. Torque performance was measured using both constant and stepwise current inputs. The coefficient of variance of the torque was successfully reduced by half due to the structural redesign. Although the time constants of the H-MRC were less than 10 ms, those of the TD-MRA were less than 20 ms under all conditions. To address the slower downward output response, we implemented an improved input method, which successfully halved the response time.

## 1. Introduction

Teleoperation systems [1,2] are installed in disaster-affected areas [3,4], underwater [5,6], and in surgical robots [7], among other delicate and socially responsible tasks. In the surgical robot, the unilateral controller [8] had been used in the most popular surgical robot, da Vinci Surgical System®, with less haptic feedback for surgeons [9]. Evidence of increasing tendencies for concomitant medical problems in laparoscopic surgeries has been reported [10], and robot-aided surgery is a solution for safe operations [11]. However, concomitant medical problems have been reported in such robot-aided surgeries due to unexpected pressure of robotic tools for target/non target organs [12]. In addition, the learning curve of robotic surgery tends to be longer than that of other methods [13]. Hence, the haptic feedback function has been heavily demanded by surgeons to enhance operational functions and reduce psychological stresses.

To address these requirements, novel teleoperation systems for surgical operations [14] incorporate haptic feedback using bilateral controllers [15,16]. These controllers support operational skills through gravity/friction compensation and haptic perception, reducing mental stress. In bilateral controllers, the operator and the remote site are referred to as the “leader” and “follower”, respectively, with their positions and reaction forces controlled to be nearly equal. However, the high reduction ratio, inertia, and friction of the leader system can deteriorate the quality of haptic feedback and the stability of position control for both systems.

To establish a high-performance force-feedback system with low inertia, high back-drivability, and high torque for direct driving, we used magnetorheological fluid (MRF) [17] as the core material for fine haptic feedback devices in surgical robot systems [18,19]. MRFs are functional fluids that exhibit rapid and reproducible rheological responses to external magnetic fields. Their use in haptic devices is justified due to their high performance in force variations.

Regarding medical application with of the MRF, Liu et al. [20] reported a review article. As reported in this article, tactile and haptic devices are important application of the MRF, and helpful to achieve leader systems with force feedback functions. For tactile device, Scilingo et al. [21] compared the compressional compliance of several MRF samples when different magnetic field intensities were applied with specimens of bovine biological tissues. Bicchi et al. [22] and Scilingo et al. [23] developed two prototypes of tactile device: Pinch Grasp and Haptic Black Box I. Later, Rizzo et al. [24] proposed Haptic Black Box II which could materialize some quasi-3D virtual objects. Liu et al. [25] also conducted preliminary research on MRF for tactile device. A single cell MRF-based tactile device equipped with two kinds of electromagnets was designed and tested. Lee et al [26] further designed a multi-cell tactile device using MRF. Tsujita et al [27] designed a tactile device containing MRF to mimic soft tissues and display cutting forces for surgical simulators. This type of devices utilized the tactile variation of the MRF through soft and hard intermediate tools, and the required size of electric magnets is relatively large. In addition, the sealing of the MRF is one of the challenges.

As another solution, haptic devices were also proposed with rotary MRF dampers, clutches, and actuators. Ahmadkhanlou et al. [28] developed a joystick-type haptic device with a rotary MRF damper for telerobotic surgery. Yin et al. [29] developed an MRF-based catheter haptic device for vascular interventional surgery. Song et al. [30] developed a haptic device using MRFs based on a planetary gear mechanism. In contrast, we have developed compact MRF clutches for haptics (H-MRCs) and installed them in the first prototype of a twin-driven MRF actuator (TD-MRA) [31]. Wellborn et al. [32] designed and developed a small-scale MR brake with a hybrid rotor combining disc and drum types for haptic applications. On the other hand, we utilized a multi-layered disc structure to achieve a miniature size of the H-MRCs. The TD-MRA successfully generated fine haptic feedback for operators.

However, a method for estimating the torque performance of the H-MRCs has not yet been reported. Therefore, we measured the rheological properties of a sample of MR fluid and developed a model based on these properties [33]. The torque performance of the H-MRC was then estimated using this model and the results from electromagnetic analyses. However, undesirable torque ripples were observed due to shaft misalignment and the low rigidity of the structure [34]. Additionally, detailed torque control performance was not evaluated from both static and dynamic current inputs. Consequently, the objective of this article is to develop a second prototype to reduce torque ripple by improving its structure and to evaluate its static and dynamic torque performance.

## 2. Structure and Design of Twin-Driven MR Fluid Actuator

### 2.1. Development of H-MRC

In this study, we developed a lightweight H-MRC (Figure 1a). In this structure (Figure 1b), multilayered discs are fixed to the input and output shafts, and the MRF fills the space between these discs. The basic structure exhibited rotational symmetry, and double bearings held the rotational shaft. A magnetic wire is wound and embedded in the magnetic core. The discs and magnetic core were made of iron, whereas the other parts were made of non-magnetic materials. The optimal design method for a similar structure was discussed in [35] and its design method were used in this device. For the optimal design, we focused on maximizing the maximum-to-minimum torque ratio under the restrictions of a compact servo amplifier (maximum current output: 2.0 A) as an easily obtainable current amplifier.

Based on the results of the optimal design, a 0.3 Nm-Class H-MRC was developed [34]. The specifications of the device are listed in Table 1. The device has two groups of rotational discs (core and casing sides). In the case of three pairs, the numbers of discs on the core and casing sides were two and three, respectively. The fluid gap is an important parameter in MRF devices. Because of the low-friction sealing, the idling torque was approximately 0.005 Nm (measured value). Low friction in the off state is an important property of fine haptic devices.

### 2.2. Basic Structure of TD-MRA and First Prototype

Figure 2 illustrates the basic concept of TD-MRA. In this structure, an input motor generates a driving torque in only one direction (e.g., CW) at a constant speed, and a pair of gears generates torque in both CW and CCW directions. In each direction, the H-MRC precisely transmitted the torque. Subsequently, the CW and CCW torques are combined using linkage or belt mechanisms. Finally, the differential torque was the output [31]. This mechanism reduces the output inertia and mechanical friction.

In [16], we developed the first prototype of a 0.3-Nm-Class TD-MRA (Figure 3). This prototype consisted of a geared alternating-current (AC) servo actuator (RSF-8B-30-F100-24B-C, Harmonic Drive Systems Inc., Osaka, Japan), a pair of flat gears, two 0.3-Nm-Class H-MRCs, a parallel link, and a rotary encoder. The AC servo actuator provided drive torque at a constant speed, and its rotation was controlled through forward and reverse motion using the pair of flat gears. However, undesirable torque ripples were observed due to shaft misalignment and the low rigidity of the structure [34].

### 2.3. Development of the Second Prototype

The structure was redesigned to improve these properties, as shown in Figure 4. The H-MRC was held with an Oldham coupling on both sides to resolve the misalignment. In addition, the attachment walls were strengthened. For the rotor resource, we used an ultrasonic motor (PSM60S-ET, Piezo Sonic Corporation, Tokyo, Japan) because of its compactness. Furthermore, a hollow-type rotary encoder was inserted between the output linkage and the H-MRC. A comparison of the dimensions of the two devices is shown in Figure 5. 

## 3. Method for Torque Control Performance Evaluation

### 3.1. Measurement Setup

Figure 6 illustrates the signal flow of the measurement system. A personal computer (PC) served as the main controller, while an NI PXIe-8821 was used as the real-time controller. They communicated via local-area network cables. The sampling time of the NI PXIe-8821 was 1 ms. The D/A board (NI PXIe-6738) was inserted into the NI PXIe-8821 to output the command values for controlling the ultrasonic motor and clutches. The output torque of the TD-MRA was measured in real time using a load cell (LMA-A-50N, Kyowa) connected through an aluminum link (length: 14 mm). The A/D board (NI PXIe-6755), also inserted into the NI PXIe-8821, read the output signal from the load cell to measure the output torque of the TD-MRA.

### 3.2. Experimental Conditions and Analytic Method

A commercially available MRF, 132DG (Lord Corp., relative density: 2.98–3.18, particle fraction: 32 vol%), was selected as a working material for the H-MRC. Its flow characteristics are summarized in our previous article [33].

We conducted two types of tests to investigate the torque performance, as follows: Static test (constant current input),Dynamic test (stepwise current input).

For the static tests, the rotational velocity of the ultrasonic motor was controlled at 10, 20, 30, and 60 rpm to study the effects of the rotational velocity. The electric current at each velocity was maintained at 0.0, 0.1, 0.2, 0.3, 0.6, 0.9, 1.0 and 1.2 A for 10 s. The average output torque and coefficient of variance (CV) were calculated for each condition. 

For the dynamic tests, the rotational velocity was controlled at 10, 30, and 60 rpm for 6 s, and the electric current for the H-MRC in CW direction was controlled from 0.0−0.5 A, 0.5−0.0 A, 0.0−1.0 A, and 1.0−0.0 A in a stepwise manner. In contrast, the electric current for the H-MRC in the CCW direction was maintained at zero under all conditions (left of Figure 7). The time constant for each condition was calculated as the representative value of the response time (right side of Figure 7). 

## 4. Results

### 4.1. Static Test

Figure 8 shows the time profiles in the static tests for input currents of 0.0, 0.1, 0.5 and 1.0 A, respectively at each rotational velocity. These results indicate a low level of torque ripple. Figure 9 shows the average torque for each condition. The horizontal and vertical axes represent rotational velocity and output torque, respectively. The output torque is slightly affected by the rotational velocity. 

### 4.2. Dynamic Test

Figure 10, Figure 11 and Figure 12 shows the step responses. The horizontal axes show the time in seconds and the vertical axes show the output torque. The current increases from an initial value to a final value of current at 3.0 s.

## 5. Discussion

### 5.1. Static Test

The coefficients of variance (CVs) were calculated from data in 2.5–7.5 s to avoid the effect of motor acceleration and deceleration and listed with the average torque in Table 2. The CVs are around 1.0% for 0.1 A input and less than 0.5% for the other conditions. As reported in [33], the CVs of the first prototype at 0.1, 0.5, and 1.0 A were 6.3%, 1.6%, and 1.1% at 60 rpm, respectively. Therefore, the second prototype successfully reduced torque ripple because of the redesign of its structure. 

### 5.2. Dynamic Test

The time constant, which is the time for the step response to reach 63.2% of its final value, is evaluated as the response time (Table 3). The step response of the first prototype was not sufficiently evaluated due to the torque ripple. However, we could successfully measure the time constant for the second prototype. Although the time constants of H-MRC were less than 10 ms [34], those of TD-MRA were more than 10 ms for almost all conditions and less than 20 ms for all conditions. This is because the actuator-level responsiveness can be affected by the response of the H-MRC and motor. The lowest rotational speed (10 rpm) resulted in the slowest response in the same-current group. The rotational speed may affect the cluster generation speed in the MRF. In addition, the downward outputs, in particular the response from 1.0–0.0 A tend to show a slower response in this actuator. The detailed reason for this is not yet clear; however, it is possible to derive it from the output properties of the ultrasonic motor and the performance of its driver circuit. 

To improve the slower tendency of the downward output, we additionally attempted an improved input method, as shown in Figure 13. In this method, the input current for the H-MRC in the CW direction was the same as that shown in Figure 7. In addition, the input current for the H-MRC in the CCW direction was controlled as pulse-wise with a pre-registered width. The height of the pulse wave was controlled such that it was equal to the step height of the input to the CW motor. In this study, the pulse width was controlled at 30 ms. The experimental results and calculated time constants for the responses are summarized in Figure 14, Figure 15 and Figure 16 and Table 4. As shown in these results, the response times were reduced by more than half. 

Goncalves and Carlson [36] estimated the response time from the dwell time of an MRF-based valve and reported that it was several hundred microseconds, which is evidence of the response time of the dynamics of MRFs. However, the response time of the MRF device depends on both pure dynamics of the MRFs and response of the magnetic coils and electric circuits of the current amplifiers used in the entire system. Therefore, using different types of the devices can possibly have non-negligible effects on the response time of the device. Detailed analyses, modeling and control on the dynamic response of the device should be discussed in future.

## 6. Conclusions

We developed several types of compact MRF clutches for haptics (H-MRCs) and applied them to a twin-driven MRF actuator (TD-MRA). The first prototype of the TD-MRA successfully generated fine haptic feedback for operators. However, undesirable torque ripples were observed due to shaft misalignment and the low rigidity of the structure. Therefore, the objective of this study was to develop a second prototype to reduce torque ripple by improving its structure and evaluating its static and dynamic torque performance. Torque performance was measured using constant and stepwise current inputs. As a result, the coefficient of variance of the torque was successfully reduced by half due to the structural redesign. Although the time constants of the H-MRC were less than 10 ms, those of the TD-MRA were less than 20 ms under all conditions. To address the slower downward output response, we implemented an improved input method, which successfully halved the response time. One of limitations of this article is a lack of experiments in long-term. Measurements for creep phenomenon and repeat accuracy should be conducted in future. 

## Figures and Tables

**Figure 1 micromachines-15-01184-f001:**
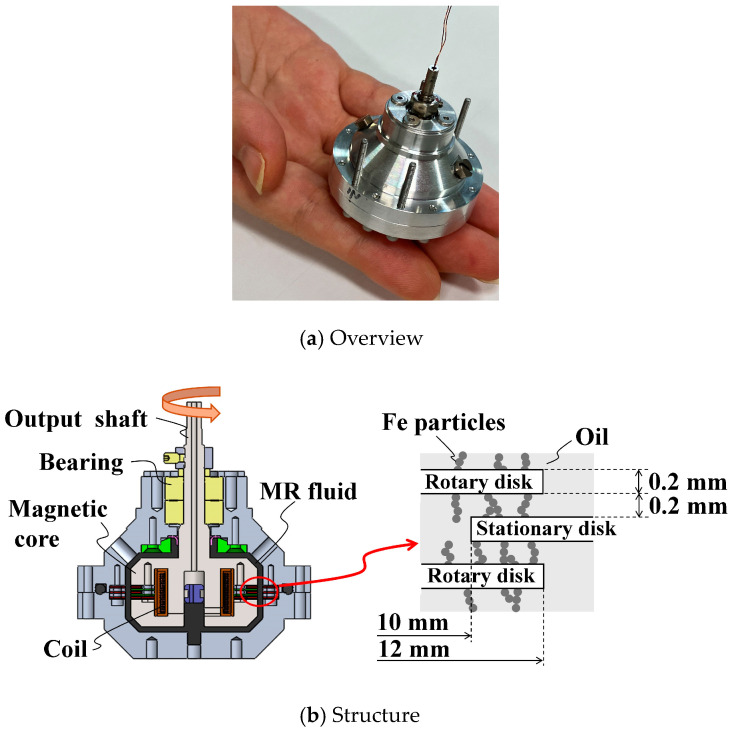
MR fluid clutch for haptics (H-MRC).

**Figure 2 micromachines-15-01184-f002:**
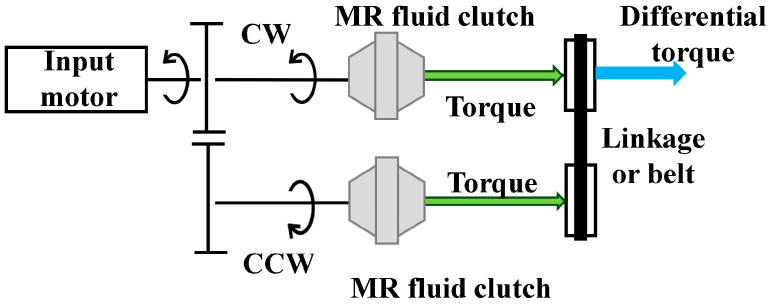
Basic structure of TD-MRA for haptics.

**Figure 3 micromachines-15-01184-f003:**
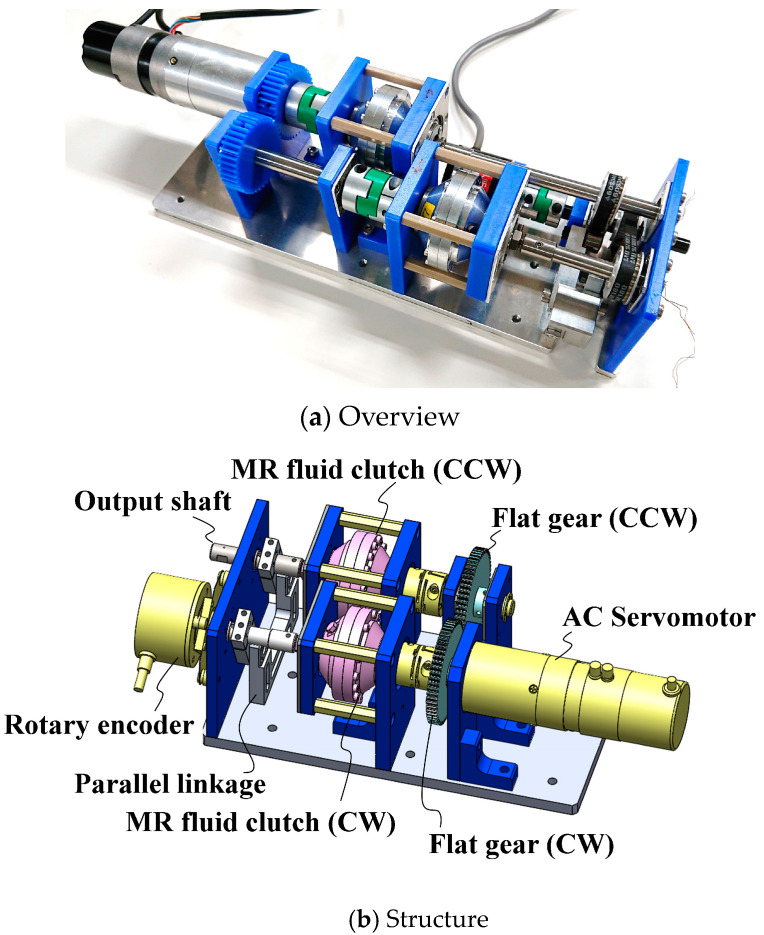
First prototype of 0.3 Nm-Class Twin-driven MR Fluid Actuator (TD-MRA 1st).

**Figure 4 micromachines-15-01184-f004:**
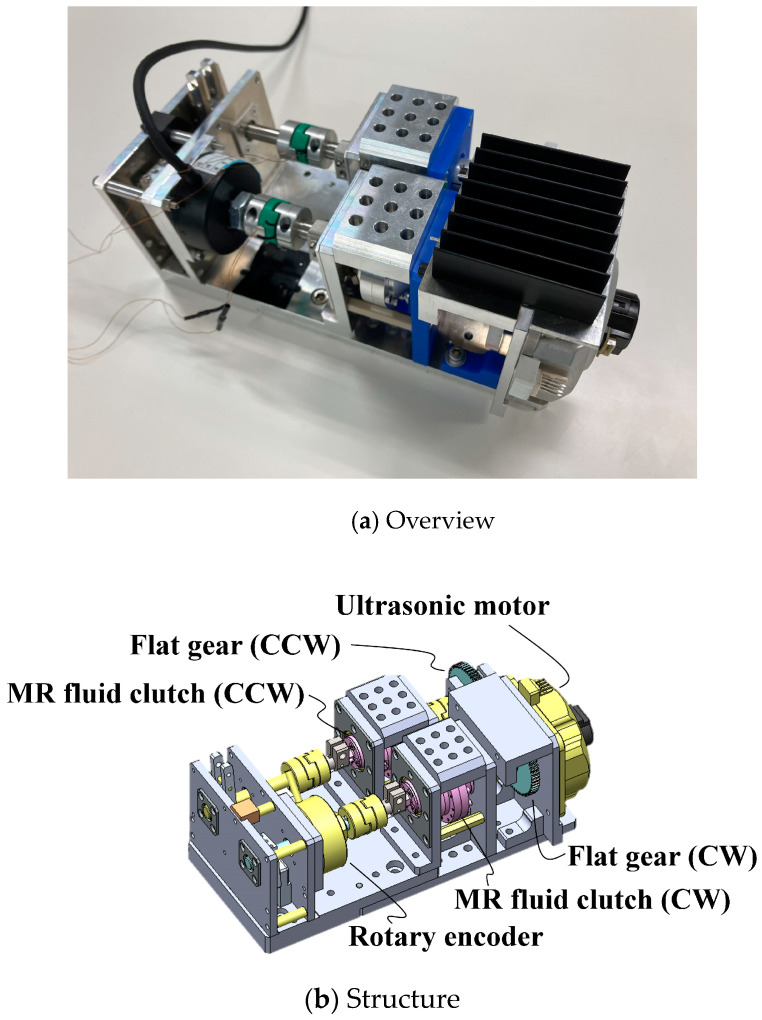
Second prototype of 0.3 Nm-Class Twin-driven MR Fluid Actuator (TD-MRA 2nd).

**Figure 5 micromachines-15-01184-f005:**
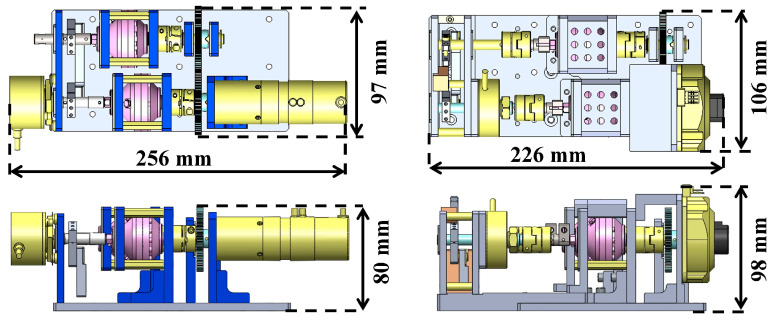
Comparison of dimension for two devices (Upper: 1^st^ prototype, lower: 2^nd^ prototype).

**Figure 6 micromachines-15-01184-f006:**
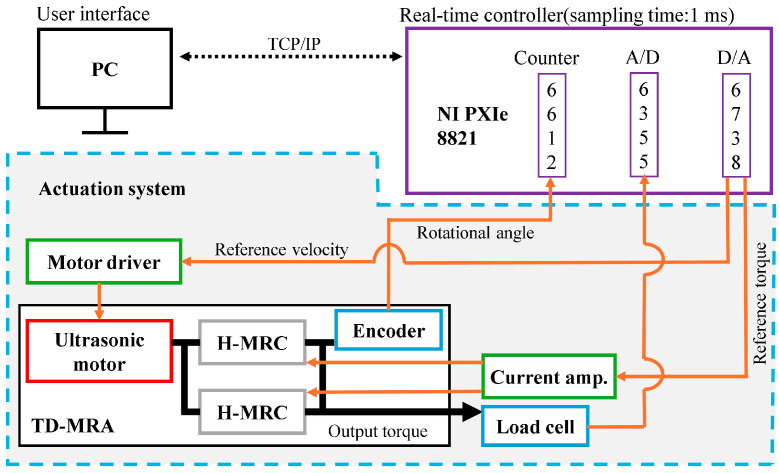
Signal diagram of measurement system.

**Figure 7 micromachines-15-01184-f007:**
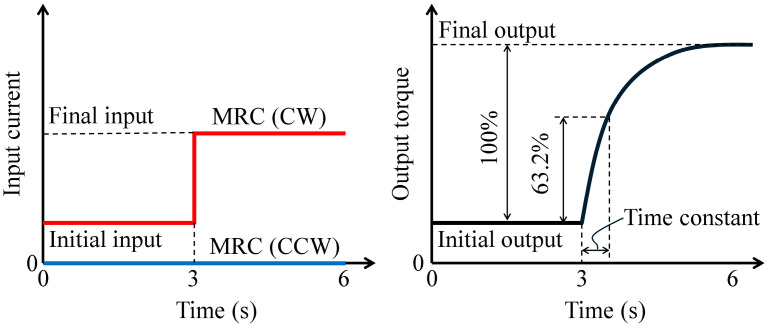
Profile of input current and definition of time constant.

**Figure 8 micromachines-15-01184-f008:**
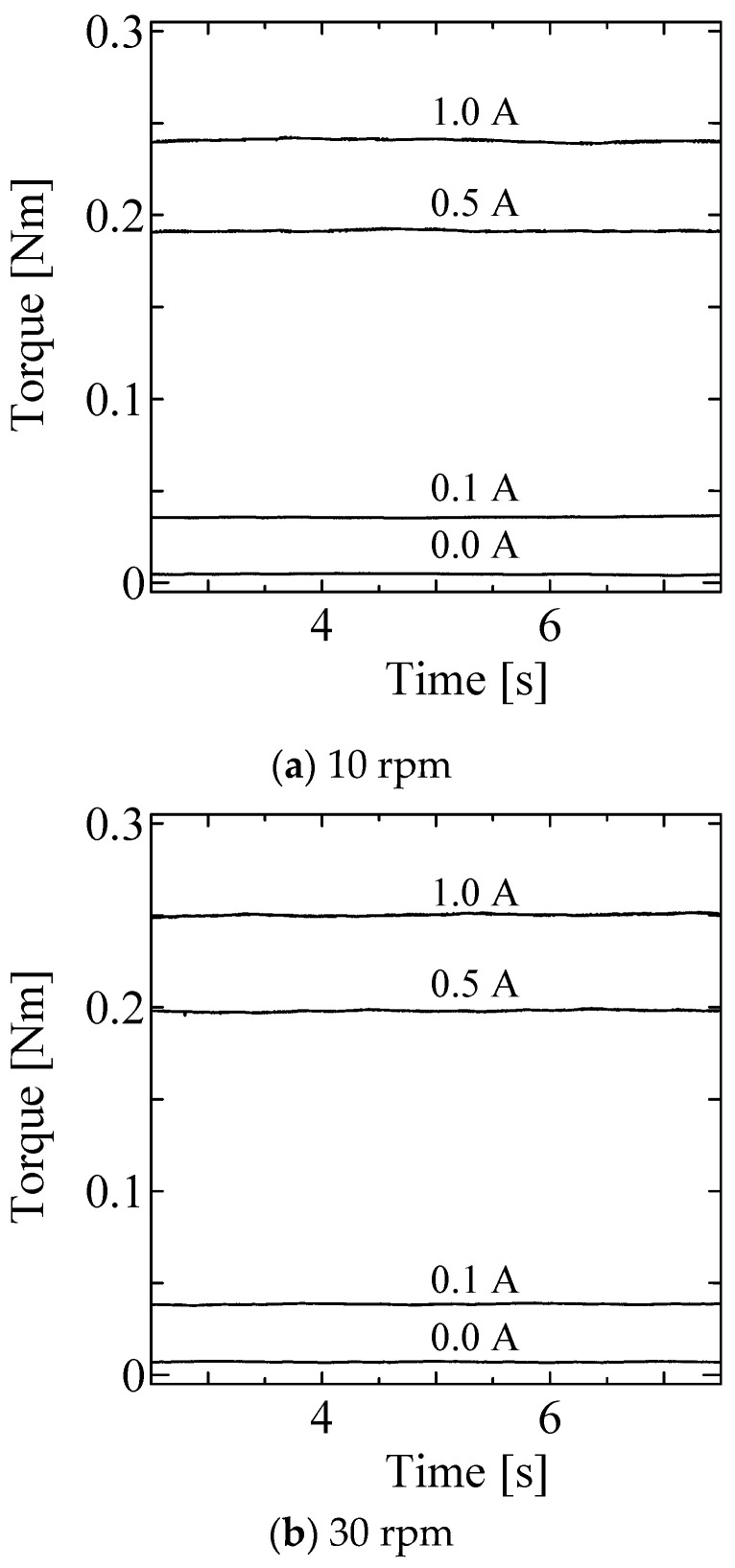
Results of static test (time profile).

**Figure 9 micromachines-15-01184-f009:**
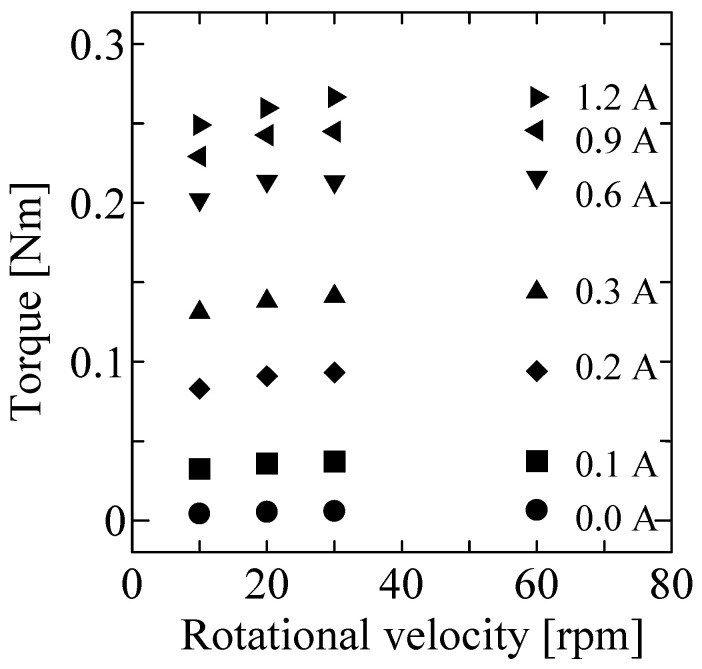
Results of static test (average values).

**Figure 10 micromachines-15-01184-f010:**
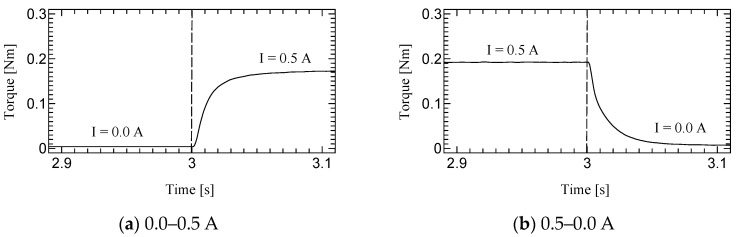
Results of Dynamic test at 10 rpm.

**Figure 11 micromachines-15-01184-f011:**
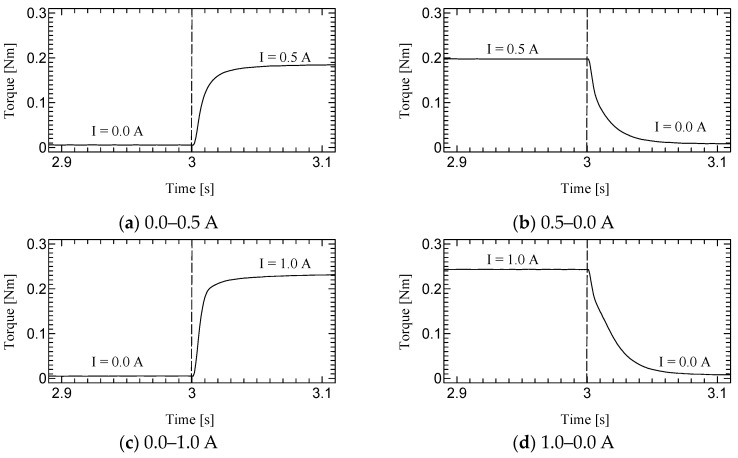
Results of Dynamic test at 30 rpm.

**Figure 12 micromachines-15-01184-f012:**
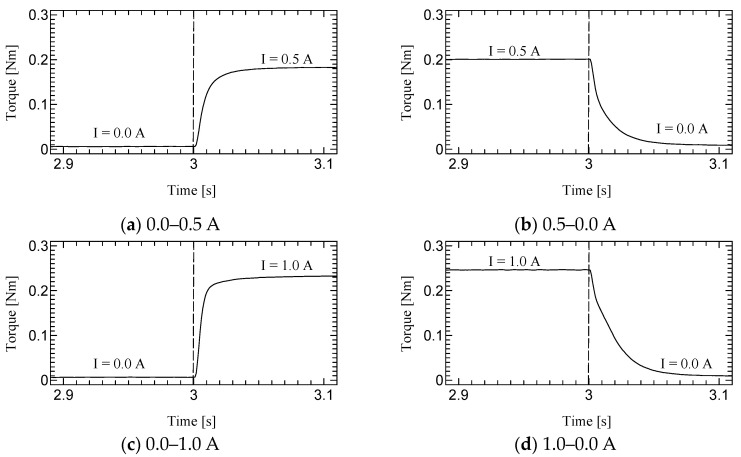
Results of Dynamic test at 60 rpm.

**Figure 13 micromachines-15-01184-f013:**
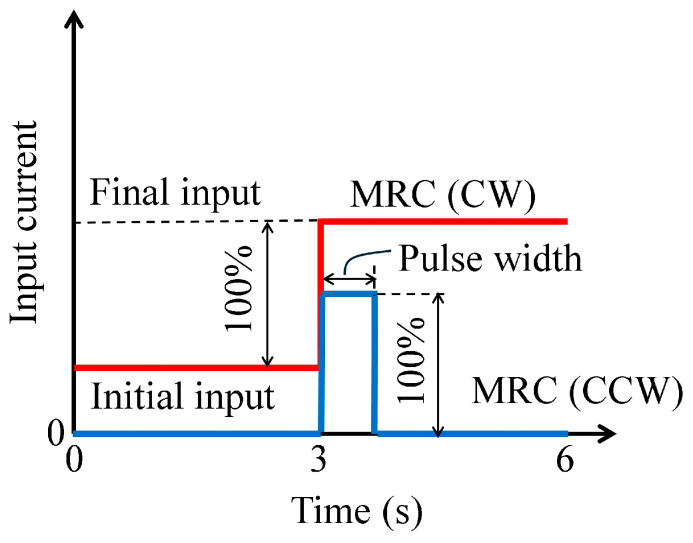
Profile of modified input current.

**Figure 14 micromachines-15-01184-f014:**
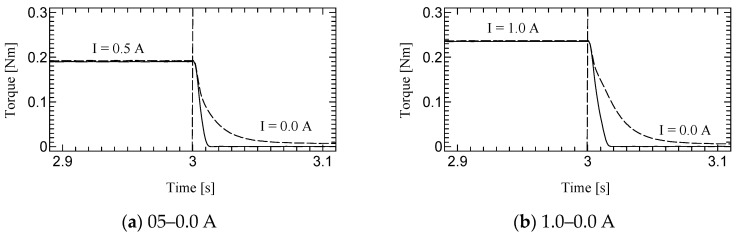
Results of Dynamic test with modified input at 10 rpm.

**Figure 15 micromachines-15-01184-f015:**
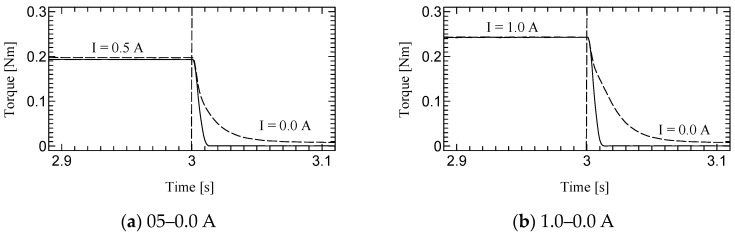
Results of Dynamic test with modified input at 30 rpm.

**Figure 16 micromachines-15-01184-f016:**
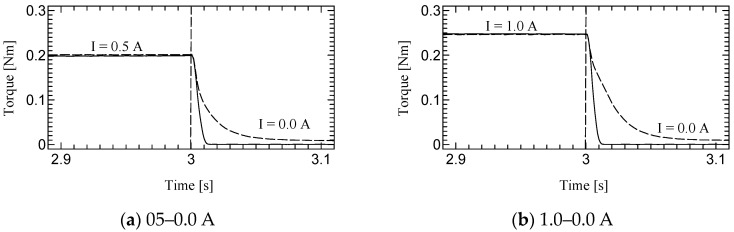
Results of Dynamic test with modified input at 60 rpm.

**Table 1 micromachines-15-01184-t001:** Specifications of H-MRC in this study.

Parameters	Values
Diameter of housing	44 mm
Height without shaft	35 mm
Pair number of discs	3
Thickness of disc	0.2 mm
Thickness of MRF layer	0.2 mm
Inner radius of disc	10 mm
Outer radius of disc	12 mm
Turning number of the coil	154
Mass with MRF	<120 g
Inertia of output part	2.47 × 10^−6^ kg·m^2^
Output torque	0.25 Nm @ 1.2 A

**Table 2 micromachines-15-01184-t002:** Coefficient of variance (CV).

Current [A]	Rot. Velocity [rpm]	Average Torque [Nm]	CV [%]
0.1	10	0.03559	1.07
	30	0.0384	1.23
	60	0.0384	1.30
0.5	10	0.190	0.27
	30	0.197	0.34
	60	0.201	0.29
1.0	10	0.241	0.34
	30	0.250	0.23
	60	0.256	0.21

**Table 3 micromachines-15-01184-t003:** Response time (time constant) for step input.

Current [A]	Rot. Velocity [rpm]	Time Constant [ms]
0.0–0.5	10	14
	30	10
	60	10
0.5–0.0	10	13
	30	12
	60	13
0.0–1.0	10	16
	30	8
	60	7
1.0–0.0	10	20
	30	20
	60	20

**Table 4 micromachines-15-01184-t004:** Response time (time constant) for modified input.

Current [A]	Rot. Velocity [rpm]	Time Constant [ms]
0.5–0.0	10	6
	30	6
	60	6
1.0–0.0	10	8
	30	6
	60	6

## Data Availability

The original contributions presented in the study are included in the article, further inquiries can be directed to the corresponding authors.

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
