# Peer review of "Development of Second Prototype of Twin-Driven Magnetorheological Fluid Actuator for Haptic Device"

_micromachines, 2024, doi:10.3390/mi15101184_

Round 1

Reviewer 1 Report

Comments and Suggestions for Authors

Paper Review

This paper proposes an improved twin driven MRF actuator, which successfully reduces the adverse torque ripple generated by the original device and shortens the response time of the device through an improved current input method. It has certain application value.

The main issues of the article are as follows:

1. The usage of words in the abstract is inappropriate, for example, in the sentence "An MRF has been utilized..." on line 9, it should be changed to "A MRF has been utilized...".

2. The grammar in the abstract is inconsistent, for example, the voice of the sentence "The objective of this study is to develop a second prototype to reduce torque ripple by improving the structure and evaluating its static and dynamic torque performance." is not consistent with the tense of the preceding and following sentences, and needs to be explained or modified.

3. The summary of keywords is not sufficient, at least four keywords should be highlighted.

4. There are multiple instances of inconsistent voice in the main text that require explanation or correction, such as the paragraphs in lines 60-61 and lines 70-71.

5. Figure 8 does not clearly show the effect of rotational speed on output torque, and Table 2 only presents the variance values. Is there a numerical table to quantitatively display the effect of rotational speed on the average torque value.

6. Table 2 only lists the torque variance value CV of the improved device, but the analysis above the table mentions the variance before the improvement, and specific values need to be listed in the table for comparison. Similarly, for the time indicators in Table 3, it is necessary to list the specific values of the pre improvement plan.

7. What are the performance parameters of the magnetorheological fluid used in this device, and what is the working channel radius of the magnetorheological fluid in the clutch device? These parameters need to be supplemented.

8. The vertical axis of Figures 7 and 13 lacks scale markings and needs to be corrected.

Comments on the Quality of English Language

The English quality of the article needs further improvement. Please refer to the first, second, and fourth items in the Paper review for specific revision suggestions.

Author Response

Thank you very much for taking the time to review this manuscript. Please find the detailed responses below and the corresponding revisions/corrections highlighted in the re-submitted files. The corresponding parts of the manuscript were colored in red. I hope this revision makes the manuscript publishable.

Reviewer 2 Report

Comments and Suggestions for Authors

This is an interesting paper on haptics using magnetorheological fluid clutch type systems. Generally well done.  However, the literature review is poorly done. There are many papers on magnetorheological clutches and haptics, even recently, e.g.

https://doi.org/10.3390/app14093697

10.1109/ICRA.2014.6906951

https://doi.org/10.1177/1045389X1562004

But, there are so many papers on this topic, and the authors have not shown how their work extends the state of art, and is novel and original. Six of the 16 references are the authors own, and the literature survey is simply inadequate, and does not do justice to the content of the paper.

The authors should comment further on their static test.  MRFs are well known to creep (deflect under constant load or torque), so does their device also creep under static load? This should be fully examined. It is not clear if 6 s is sufficient for such an assessment for constant RPM.

Comments on the Quality of English Language

Minor English editing is needed.

Author Response

(The authors gave the same response as above.)
